# Energy-harnessing problem solving of primordial life: Modeling the emergence of catalytic host-nested parasite life cycles

**Bernard Conrad**[1]*, **Christian Iseli**[2], **Magnus Pirovino**[3]

**1** Genesupport, Lausanne, Switzerland, **2** Bioinformatics Competence Center, EPFL and Unil, Lausanne, Switzerland, **3** OPIRO Consulting Ltd, Triesen, Principality of Liechtenstein

* bernard.conrad@genesupport.ch

## Abstract

All life forms on earth ultimately descended from a primordial population dubbed the last universal common ancestor or LUCA via Darwinian evolution. Extant living systems share two salient functional features, a metabolism extracting and transforming energy required for survival, and an evolvable, informational polymer–the genome–conferring heredity. Genome replication invariably generates essential and ubiquitous genetic parasites. Here we model the energetic, replicative conditions of LUCA-like organisms and their parasites, as well as adaptive problem solving of host-parasite pairs. We show using an adapted Lotka-Volterra frame-work that three host-parasite pairs–individually a unit of a host and a parasite that is itself parasitized, therefore a nested parasite pair–are sufficient for robust and stable homeostasis, forming a life cycle. This nested parasitism model includes competition and habitat restriction. Its catalytic life cycle efficiently captures, channels and transforms energy, enabling dynamic host survival and adaptation. We propose a Malthusian fitness model for a quasispecies evolving through a host-nested parasite life cycle with two core features, rapid replacement of degenerate parasites and increasing evolutionary stability of host-nested parasite units from one to three pairs.

**Data Availability Statement:** https://github.com/BICC-UNIL-EPFL/CatalyticHostParasiteCycles [github.com] https://doi.org/10.5281/zenodo.7261272 [doi.org].

## Introduction

One common definition of life posits that it represents a self-sustained chemical system capable of evolution [1]. While there is no general consensus on what life is and how it originated [2], it is widely appreciated that all life forms on earth ultimately descended from a theorized, primordial population named LUCA via Darwinian evolution [3, 4]. Extant living systems share two key functional components, a metabolism extracting and transforming energy required for survival, and an evolvable, informational polymer conferring heredity, i.e. the genome. Two competing–although not mutually exclusive–concepts stipulate that either primitive self-replicators (genetics-first) or self-reproducing and evolving proto-metabolic networks (metabolism-first) were critical at the origin of life [5, 6].

**Funding:** The authors received no specific funding for this work.

**Competing interests:** The authors have declared that no competing interests exist.

Genetic parasites are an integral feature of genome replication, and constitute the most abundant and diverse biological entity on earth [7]. Arguably the emergence and persistence of parasites is unavoidable, since parasite-free states are evolutionary unstable, and microbial populations cannot simultaneously clear parasites yet escape extinction through Muller's ratchet [8, 9]. They parasitize all cellular life forms including LUCA, with the possible exception of certain obligate intracellular bacteria with a reduced genome leading a parasitic life style [10]. Attempts at reconstructing the LUCA parasitome revealed a remarkably large virome comprising the extant viruses of bacteria and archaea [11]. Intriguingly, ancient bacterial symbionts undergoing massive genome erosion repeatedly experienced extinction and replacement by pathogenic microbes [12]. Under experimental conditions the process of how free-living micro-organisms become symbionts is remarkably rapid and requires one single mutation [13]. Collectively, this suggests that evolution of life essentially is host-parasite coevolution [7]. Indeed, host-parasite coevolution is considered among the dominant drivers of biological diversity over the last 3.5 billion years [14]. It further implies a life cycle of host-parasite interactions that evolve along a temporal trajectory of competition, cooperation, and replacement by fresh parasites once the ancient symbiont genome is eroded [15], (S1 and S2 Figs). As a corollary, a host exposed to a given parasite encounters a parasite that is itself parasitized (hyperparasite), forming what we henceforth call a host-nested parasite unit. Hyperparasite infections variably attenuate or increase a pathogen's virulence, steering virulence away from the optimal evolutionary stable strategy thought to represent a trade-off between transmission and host-damage [16]. Unsurprisingly, the evolutionary outcome of an individual host-nested parasite unit is complex and highly context-dependent, sometimes conferring evolutionary advantages to both, the host and the parasite, for instance via superinfection resistance and/or resistance to other parasites [17], or via spread of virulence factors [18].

A host therefore is a dynamic host-nested parasite unit with multiple, changing partners that evolves along a temporal axis implying competition and cooperation. We set out to implement this paradigm for a LUCA-like founder population, including its precursors, using the Lotka-Volterra (LV) equations. Lotka [19] and Volterra [20] independently modelled prey-predator interactions, both concluding that populations would oscillate because of this interaction. The respective equations became known as LV equations, which reduced the then popular thinking of complex food chains to the interaction of only two species controlling each other in a cyclic manner [21]. Intra- and interspecies competition is one of strong features of these equations. For instance, Campbell's model integrated intraspecific competition, formally represented by a habitat restriction term [22]. He further extended it to interspecific competition, arguing that based on the competitive exclusion principle [23, 24], only the fittest host is considered "reasoning in this way, one can conclude that a phage can survive without destroying its host simply by having chosen the best-adapted from a group of competing hosts" [22].

Collectively, we chose the LV equations, i) because they cover the widest scope [25]; ii) the extensions by Campbell integrate both intra- and interspecies competition; and iii) empirical data show that these models accurately reflect for instance bacteriophage interactions in aquatic systems [26], and both oceanic and lake virioplankton dynamics are consistent with LV [27, 28], conditions akin to the intention of our model.

## Materials and methods

### Definitions and glossary

**Fitness.** Let $P$ be a quasispecies with $N$ subpopulations $P = (P_1, P_2, \ldots, P_N)^T$. The symbol $^T$ denotes the transposed of a vector or a matrix. $m = m_i$ denotes the (Malthusian) fitness: $\frac{d}{dt}P_i = \dot{P}_i = m_i P_i, i = 1, \ldots, N. p_i = \frac{p_i}{\sum p_j}$ denotes the frequencies of the subpopulations

$p_i$ within quasi-species. $\overline{m(t)} = \overline{m} = \sum m_i p_i = m^T p$ is the average fitness and $var(m(t)) = \overline{m(t)^2} - \overline{m(t)}^2$ is the fitness variance of the quasispecies at time $t$.

**Fitness of parasites.** The term $a_1 > 0$ is the fitness of prokaryote $X$ within its host habitat in the absence of phage $Y$. $a_2 > 0$ is the fitness decrease of the prokaryote $X$ infected by phage $Y$. The term $b_1 < 0$ is the negative fitness of phage $Y$ in the absence of nutrition, i.e. prokaryote $X$, within the host habitat. $b_2 > 0$ is the fitness increase of phage $Y$ after having infected $X$.

**Virulence.** Virulence is defined in its broadest sense, i.e. the cost imposed to the host as a consequence of infection, which translates into a reduction in host fitness due to the infection [29]. A parasite is said to be virulent in its host if it is actively reproducing. We assign values between 0 and 1 to the term virulence, virulence $v = 1$ if the parasite is maximally active and reproducing, and virulence $v = 0$ if the parasite is inactive and silenced.

**Energy.** At least two energy forms exist to fight against the fitness decay of a quasispecies, $E_1$ is a form of immediately available energy (operating power, nutrition) that the quasispecies can use to directly fight erosion, e.g. induced by mutation genetic drift [30] environmental changes and for reproduction. $E_2$ is a form of stored, excess-energy $E_1$, which in addition has an isolating effect towards the quasispecies' inner and outer environment. This isolating effect partially neu-tralizes fitness erosion. Fitness increases/decreases as the level of both types of energy in-creases/decreases.

**Catalytic principle.** A given quasispecies, as is true for all living systems, is a thermody-namic entity [31] therefore particularly subject to the second law of thermodynamics, and more generally also to all other laws of physics. For that reason, in order to increase fitness, a quasispecies can only codify and enhance processes that are physically already in place. Any quasispecies-specific metabolism has to be based on (catalytic) enhancement of transforming or consuming an available energy source, such as sunlight; by the second law of thermodynam-ics a quasispecies is subject to physical decay that increases its fitness variance. Therefore, any quasispecies-specific adaptation has to go through a (catalytic) enhancement of this decay pro-cess always accompanied by an increased fitness variance (see evolutionary problem solving).

**Evolutionary problem solving.** A quasispecies faces an evolutionary problem (i.e. muta-tion, genetic drift, environmental change) when the range of Malthusian fitness values for all its subpopulations $P_i$ is negative ($m_i < 0$, $i = 1, \ldots, N$). For evolutionary problems we use a modified formalism based on classical problem-solving in mathematics [32], namely i) a prob-lem measurement phase ("under-standing the problem"), ii) a problem resolution phase (PRP; "devising and carrying out a plan"), and iii) a fine-tuning phase ("looking back").

Applied to our model, evolutionary problem solving can be contextualized as follows. Fit-ness decay, e.g. due to mutation and genetic drift, not only contributes to intensify the evolu-tionary problem, but also contributes to its resolution (PRP). Commonly, the positive effect exerted by fitness erosion via increased variance is rather weak, compared to its negative, destructive effect. Parasites typically amplify the PRP process by accentuating destruction and in-creasing the fitness variance (catalytic principle). If this amplification is strong enough–such that the right tail of the quasispecies' fitness distribution shifts well into the positive terri-tory–then the negative, destructive effect becomes irrelevant for the evolutionary survival of the quasispecies.

If parasites survive and preserve their ability to amplify the PRP process, they then also cod-ify it, i.e. they are stable information carriers of the PRP amplification process. Provided that the PRP–via broadening of fitness variance–is amplified strongly enough, moving some host members of the quasispecies into the positive fitness range, implies that the following must have happened. Some of the parasites-hyperparasites became symbiotic and therefore produce a positive fitness increase that is exerted both on themselves and on their hosts. This subgroup

of now symbiotic parasites-hyperparasites carries the essential information of the PRP, specifically reflecting the fitness-problem the quasispecies has been facing.

## The model

**The LV-host-nested parasite equations with habitat restriction (LVhpr) model.** We consider two populations, $X$ the host and $Y$ the phage. The basic model is represented by the logistic LV equations by Allan Campbell [22] describing the dynamics between the host $X$ and its phage $Y$. Here we use a simplified version of the original Campbell-Eq [22]:

$$\frac{dX}{dt} = X(a_1\left(1 - \frac{X}{L}\right) - a_2 Y) \tag{1.1}$$

$$\frac{dY}{dt} = Y(-b_1 + b_2 X) \tag{1.2}$$

where $a_1 > 0$ is the growth rate of $X$ in absence of phage $Y$, and $b_1 > 0$ is the death rate of phage $Y$ in absence of host $X$. $L$ is the maximum population of $X$, the underlying habitat is able to sustainably feed.

As a result, we obtain the equilibrium populations $\overline{X}$ and $\overline{Y}$ as follows:

$$\overline{X} = \frac{b_1}{b_2}$$

$$\overline{Y} = \frac{a_1}{a_2}(1 - \frac{b_1}{b_2 L})$$

Allan Campbell also analyses the effect of a competing host $X_{competing}$ not having a phage on the replication dynamics. In the absence of a phage "the faster growing will always displace the slower" [22]. "When the host for the phage has a selective advantage, even a very slight one, the competitor has no effect on the final density of the host bacterium [22]. "Reasoning in this way, one can conclude that a phage can survive without destroying its host simply by having chosen the best-adapted from a group of competing hosts" [22]. For this reason, we only model one, i.e. the fittest host population per phage, not extending the model to further intraspecific competition.

We further extend the simplified Campbell host-phage model ((1.1),(1.2)) into a situation, where both the host and its phage are parasites of a common overall host. In what follows, we call this the overall host population $H$. The old host population $X$ for the phage $Y$ is now called the parasite population, and the phage population $Y$ remains unchanged (hyperparasite). Again, we start analyzing the reproduction dynamics of parasite $X$ and its hyperparasite $Y$. In principle, $X$ and $Y$ obey the same reproduction dynamics as in the simplified Campbell-Model ((1.1),(1.2)), $L$ being the maximum population for parasites $X$. Since both $X$ and $Y$ are parasites hosted by $H$, we can view $H$ essentially as being the habitat for $X$ and $Y$. Thus, we write

$$L = \kappa\, H$$

in (1.1),(1.2), for some positive constant $\kappa$, which we call here the parasite capacity of the hosts in a given habitat. This is valid as long as the host population $H$ is relatively stable, while $X$ and $Y$ fluctuate around their respective equilibria. As just stated, not only $X$, but also $Y$ is hosted by $H$. With this in mind, it is reasonable to assume that phages $Y$ are more efficient than $X$ using the resources of $H$, say by a factor $q > 1$. Consequently, Eq (1.1) can be rewritten to (1.1*) as

follows:

$$\frac{dX}{dt} X\left(a_1\left(1 - \frac{(X + Y/q)}{\kappa\, H}\right) - a_2 Y\right) \tag{1.1*}$$

as long as $H$ is a relatively stable population compared to the velocity of the $X$ and $Y$ repro-duction cycles.

Additionally, let us assume that the parameters $a_2$ and $b_2$, in the adjusted Campbell-LV Eqs (1.1*) and (1.2) meet the following conditions:

$$a_2 XY = a_0\, p(X, Y) \text{ and } b_2 XY = b_0\, p(X, Y) \tag{2.1}$$

for some positive constants $a_0$ and $b_0$ This assumption is based on the fact that both, the reduction of parasite growth $-a_2 Y$, in Eq (1.1*), and the fitness increase of hyperparasites, $b_2 X$ in Eq (1.2) are related to the to the probability of encounter $p\,(X,Y)$ between parasites and hyperparasites. Following this line of reasoning, we further postulate that the probability $p\,(X,Y)$ of an encounter between a parasite and its hyperparasite–within unit time–can be written as follows:

$$p(X, Y) = \sigma \frac{X\,Y}{N} = \sigma \frac{X\,Y}{H} \tag{3.1}$$

$\sigma$ is some positive constant, which we call here the encounter constant of parasites and hyperparasites. $N$ is the total number of different loci where parasites and hyperparasite may potentially meet. Thus $N$ represents the size of the habitat where parasites and hyperparasite live and therefore we can conclude $N = const \cdot H$ and write for simplicity $N = H$.

Combining Eqs (2.1) with (3.1) yields:

$$a_2 = \frac{a_0\, \sigma}{H} \text{ and } b_2 = \frac{b_0\, \sigma}{H}$$

and thus, Eqs (1.1*) and (1.2) can be rewritten as follows:

$$\frac{dX}{dt} = X\left(a_1\left(1 - \frac{(X + Y/q)}{\kappa\, H}\right) - \frac{a_0\, \sigma}{H} Y\right) \tag{1.1**}$$

$$\frac{dY}{dt} = Y\left(-b_1 + \frac{b_0\, \sigma}{H} X\right) \tag{1.2**}$$

yielding the new equilibria $\overline{X}$ and $\overline{Y}$, for parasites and hyperparasites, being now linear functions of their host population $H$:

$$\overline{X} = \frac{b_1}{b_{0\sigma}} H$$

$$\overline{Y} = \frac{a_1(b_0\sigma - b_1/\kappa)q}{(q\, a_0\sigma + a_1/\kappa)b_0\sigma} H$$

Thus, both equilibria populations of parasites, $\overline{X}$ and hyperparasite $\overline{Y}$ are proportional to $H$, the population size of their common host. We call the extension $(1.1^{**})(1.2^{**})$ of the simplified Campbell version of the LV-host-nested parasite equations LVhpr model for the dynamic para-sites $X$ and hyperparasites $Y$ given a host population $H$. In the LVhpr model equilibrium populations $\overline{X}$ and $\overline{Y}$ can be viewed as being synchronized with the host population $H$.

As discussed, our justification for the use of the LVhpr Eqs (1.1**)(1.2**) here is: if the change in total population size of the host $H$ is slow with respect to time, compared to the growth and decay dynamics of the parasite (LUCA- or pre-LUCA-like organism) and hyperparasite (phage) populations $X$ and $Y$, then the host population $H$ can be used as a parameter, i.e. as the size of habitat in the LVhpr model (1.1**)(1.2**).

**Continuous and smooth virulence interpolation.** In order to model a continuous increase of the virulence function from 0 to 1 (an analogous formula applies for the step from 1 to 0) within $t_0$ and $t_0 + \Delta t$ we propose a smooth-step function as described below.

$$v(t, t_0, \Delta t) = smoothstep(t, t_0, \Delta t) = \begin{cases} 0 & t \leq t_0 \\ 3\left(\dfrac{t - t_0}{\Delta t}\right)^2 - 2\left(\dfrac{t - t_0}{\Delta t}\right)^3 & t_0 \leq t \leq t_0 + \Delta t \\ 1 & t_0 + \Delta t \leq t \end{cases}$$

**One energy, one host-nested parasite unit homeostasis LVhpr model.** Virulence $v$ (for simplicity modeled here with a discontinuous step function) of prokaryote $X$ and its phage $Y$, in dependence of the energy level $E$ and their host population $H$:

$$v = \begin{cases} 1, & \text{if } E \leq E^{lower} \\ 1, & \text{if } E^{lower} \leq E \leq E^{upper} \text{ and } E \text{ rises from below } E^{lower} \\ 0, & \text{if } E \geq E^{upper} \\ 0, & \text{if } E^{lower} \leq E \leq E^{upper} \text{ and } E \text{ falls from above } E^{upper} \end{cases}$$

The LVhpr model of prokaryote $X$ and phage $Y$ under virulence $v$, with respect to a relatively stable host population $H$:

$$\dot{X} = v X \left( a_1 \left( 1 - \frac{\left(X + Y/q\right)}{\kappa H} \right) - \frac{a_0 \sigma}{H} Y \right),$$

$$\dot{Y} = v Y \left( -b_1 + \frac{b_0 \sigma}{H} X \right),$$

with equilibria $\overline{X} = \frac{b_1}{b_0 \sigma} H$ and $\overline{Y} = \frac{(b_0 \sigma - b_1/\kappa) a_1 q}{(q a_0 \sigma + a_1 \kappa) b_0 \sigma} H$,

where $a_i > 0$, $b_i > 0$, $i = 0,1$, and, as defined in (1.1**), (1.2**):

$k$ is the parasite capacity of the hosts,

$\sigma$ the encounter constant between parasites and hyperparasites, and

$q$ the efficiency quotient of hyperparasites versus parasite in using the host as a habitat.

Energy consumption and production differential equation: $\dot{E} = -aH - \beta v X + \gamma v Y$, where

$\alpha > 0$ is the rate of permanent energy consumption of the host-parasite unit system $H$,

$\beta > 0$ is the rate of energy consumption of parasite $X$, when virulent,

$\gamma > 0$ is the rate of catalytic energy production of hyperparasite $Y$, when virulent.

**Two energies, three host-nested parasite units homeostasis LVhpr model.** For a quasi-species system are given two energy levels (in each host) $(E_1, E_2)$ and three parasite-hyperparasite pairs $(X_1, Y_1), (X_2, Y_2), (X_2, Y_2), (X_1, Y_2)$. In our model realization, the virulence of the pair $(X_1, Y_1)$ is regulated by $E_1$ (virulence $v_{E_1}$) and the virulence of the pairs $(X_2, Y_2), (X_3, Y_3)$ are regulated by $E_2$ (virulence $v_{E_2}$)

Virulence $v_{E_1}$ of parasite $X_1$ and its hyperparasite $Y_1$, in dependence of energy level $E_1$:

$$v_{E_1} = \begin{cases} 1, & \text{if } E_1 \leq E_1^{lower}, \text{ or if } E_1^{lower} \leq E_1 \leq E_1^{upper} \text{ and } E_1 \text{ rises from below } E_1^{lower} \\ 0, & \text{if } E_1 \leq E_1^{upper}, \text{ or if } E_1^{lower} \leq E_1 \leq E_1^{upper} \text{ and } E_1 \text{ falls from above } E_1^{upper} \end{cases}$$

Virulence $v_{E_2}$ of parasite-hyperparasite pairs $(X_2, Y_2)$, $(X_3, Y_3)$, in dependence of energy level $E_2$:

$$v_{E_2} = \begin{cases} 1, & \text{if } E_2 \leq E_2^{lower}, \text{ or if } E_2^{lower} \leq E_2 \leq E_2^{upper} \text{ and } E_2 \text{ rises from below } E_2^{lower} \\ 0, & \text{if } E_2 \leq E_2^{upper}, \text{ or if } E_2^{lower} \leq E_2 \leq E_2^{upper} \text{ and } E_2 \text{ falls from above } E_2^{upper} \end{cases}$$

The LVhpr equations of the parasite-hyperparasite pairs $(X_1, Y_1)$, $(X_2, Y_2)$, $(X_3, Y_3)$:

$$\dot{X} = \left( a_1 \left( 1 - \frac{\left( \sum X_i + \frac{1}{q} \sum Y_i \right)}{\kappa H} \right) - \frac{a_0 \, \sigma}{H} Y_1 \right) v_{E_1} X_1,$$

$$\dot{Y}_1 = \left( -b_1 + \frac{b_0 \, \sigma}{H} X_1 \right) v_{E_1} Y_1,$$

where $a_i > 0$, $bi > 0$, i = 1,2.

$$\dot{X}_2 = \left( c_1 \left( 1 - \frac{\left( \sum X_i + \frac{1}{q} \sum Y_i \right)}{\kappa H} \right) - \frac{c_0 \, \sigma}{H} Y_2 \right) v_{E_2} X_2,$$

$$\dot{Y}_2 = \left( -d_1 + \frac{d_0 \, \sigma}{H} X_2 \right) v_{E_2} Y_2,$$

where $c_i > 0$, $d_i > 0$, i = 1,2.

$$\dot{X}_3 = \left( e_1 \left( 1 - \frac{\left( \sum X_i + \frac{1}{q} \sum Y_1 \right)}{\kappa H} \right) - \frac{e_0 \, \sigma}{H} Y_3 \right) v_{E_2} X_3,$$

$$\dot{Y}_3 = \left( -f_1 + \frac{f_0 \, \sigma}{H} X_3 \right) v_{E_2} Y_3,$$

where $e_i > 0$, $f_i > 0$, i = 1,2.

Here again, the following constants are defined as in $(1.1^{**})$, $(1.2^{**})$

$\kappa$ is the parasite capacity of the host,

$\sigma$ the encounter constant between parasites and hyperparasites, and

$q$ the efficiency quotient of hyperparasites versus parasite in using the host as a habitat.

Energy consumption and production differential equations

$$\dot{E}_1 = -\alpha_1 H + \gamma_1 \theta^{-1} v_{E_1} Y_1 + \gamma_2 v_{E_2} Y_2 - \gamma_3 v_{E_2} Y_3,$$

and

$$\dot{E}_2 = -\alpha_2 E_2 + \gamma_1 v_{E_1} Y_1 + \gamma_3 \theta v_{E_2} Y_3,$$

where

$\alpha_1 > 0$ is the rate of permanent energy $E_1$ consumption of the (host-nested parasite) system $H$.

$\alpha_2 > 0$ is the rate of permanent decay of isolation material.

$\gamma_1 > 0$ is the rate of energy $E_1$ production of hyperparasite $Y_1$ (transforming $E_2$ into $E_1$), when virulent.

$\gamma_2 > 0$ is the rate of energy $E_1$ production of hyperparasite $Y_2$ (catalyzing new energy $E_1$), when virulent.

$\theta > 0$ is the rate of energy-conversion $E_1$ to $E_2$, typically $E_2 = \theta E_1$, with $0 < \theta \ll 1$.

$\gamma_3 > 0$ is the rate of energy $E_2$ production of hyperparasite $Y_3$, (transforming $E_1$ into $E_2$), when virulent.

Thus, in this model the nested parasite-pairs are information carriers for the follow processes

$(X_1, Y_1)$ transforms $E_2 \rightarrow E_1$:　　　　　　Process 1 «problem measurement phase»

$(X_2, Y_2)$ catalyzes new $E_1$, *low* $E_1 \rightarrow$ *high* $E_1$ : Process 2 «problem resolution phase»

　　　　　　　　　　　　　　　　(establish a «metabolism»).

$(X_3, Y_3)$ transforms $E_1 \rightarrow E_2$:　　　　　　Process 3 «fine-tuning phase».

## Results

We built our model by choosing five main concepts to converge, first, the fact that parasitism is an essential, constantly inherent feature of living organisms, and that a host infected by a parasite reflects on average a nested parasitism (hyperparasitism); second, a quasispecies exposed to fitness decay is offered the problem-solving opportunity to recruit new symbionts from the pool of free-living microorganisms, a very rapid process that requires a single or only few mutations. This phenomenon may well predate modern genomes and also apply to LUCA- and pre-LUCA-like organisms; third, evolutionary problem-solving materializes in a three step-mode formally involving i) a problem measurement phase ("understanding the problem"), ii) a problem resolution phase or PRP; "devising and carrying out a plan"), and iii) a fine-tuning phase ("looking back"); fourth, energy is the quintessential fuel driving evolution, and newly recruit-ted, free-living nested parasites are the key catalysts propelling energy-trans-formation during evolutionary problem-solving; fifth, an extended LV-framework is used to capture evolving host-nested parasites in a setting that includes competition and habitat-restriction (LVhpr model). Inevitably, the LVhpr model incrementally increases the number of host-nested parasite pairs during evolutionary problem solving until fine-tuning is fully implemented.

Living systems are thermodynamic entities [31], sometimes referred to as energy harness-ing device making copies of itself [33]. Boltzmann pointed out that "the fundamental object of contention in the life-struggle, in the evolution of the organic world, is available energy, and Lotka stated that "in the struggle for existence, the advantage must go to those organisms whose energy-capturing devices are most efficient in directing available energy into channels favorable to the preservation of the species" [34].

For these reasons, we modelled virulence in relation to energy. In what follows we will show in a gradual process that two energies and three trios of host-nested parasite units are re-quired to attain robust homeostasis. The host is modelled intrinsically, trios may represent temp-orally successive states of the same unit (S1 and S2 Figs). At least two energy forms counter the natural fitness decay related to mutation, genetic drift [30] and environmental changes; those energies are also required for reproduction. Both can be viewed as proxies for the Malthusian fitness of the quasispecies. We based our simulation on a simple energy-viru-lence relation (Fig 1A). Both energies, $E = E_1$ and $E = E_2$, have two levels through which they

regulate the virulence of parasites sensitive to them. If $E_1$ is abundant ($E_1 > E_1^{upper}$), the virulence of ($E_1$– sensitive) parasites is zero: $v = 0$. If $E_1$ declines, the parasite remains dormant ($v = 0$), until energy drops below a lower level $E_1 < E_1^{lower}$. This allows the parasite to be reactivated and to acquire maximal virulence ($v = 1$). If in a second time point the host's energy level rises again, the parasite remains virulent until $E_1$ reaches the upper level again, which is the threshold allowing the host to tame all of its $E_1$-sensitive parasites ($v = 0$), and so on.

In practice, a continuous virulence function would be more realistic. In order to model a continuous and smooth interpolation of the virulence function from 0 to 1 within $t_0$ and $t_0 + \Delta t$, we implemented a smooth-step function as described (Materials and Methods). As can be seen in Fig 1B, this continuously and smoothly interpolated virulence step-function does indeed make the model more realistic. In conclusion, for the scope of fundamental considerations of host-nested parasite interactions that we aim at here, the simple non-interpolated virulence step-function as used in Figs 1A–3 fully exhibits all the relevant properties under consideration.

In Fig 2A we extend this model by adding a hyperparasite (viruses e.g. bacteriophages) to the parasite (LUCA- or pre-LUCA-like prokaryote e.g. bacteria or archaea) to form the host-nested parasite unit, and limit it to one single energy ($E = E_1$). This yields the "one energy, one host-nested parasite homeostasis model". This simple model leads to an initial level of homeostasis–in the host or more generally in the whole quasispecies–as long as enough new operating power $E$ can be catalyzed by some symbiotic host-nested parasite unit (prokaryote $X$ and virus $Y$), and as long as the system does not skip into a problematic zone.

The population size value was set to $H = 4$ population units, which allows the equilibrium populations of parasites-hyperparasites to be stably achieved in a short time, by virtue of the logistic maximum habitat restriction stemming from the host population $H$. A side effect of this habitat restriction is that inter- and intraspecific competition of parasites is also covered by the model (Fig 2A).

In Fig 2A the hyperparasite (phage $Y$) alone catalyses new energy used as operating power for the host-nested parasite unit system through the equation $\dot{E} = -\alpha H - \beta v X + \gamma v Y$, where

$\alpha > 0$ is the rate of permanent energy consumption of the host-nested parasite unit system $H$,

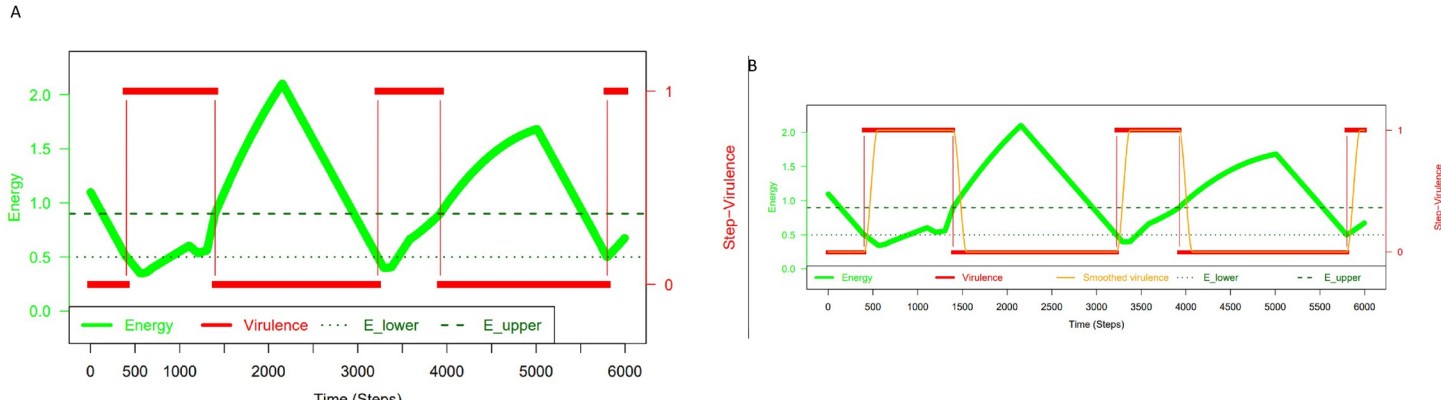

**Fig 1. Host-parasite energy-virulence model.** (A) Illustrates the relationship between the energy level $E = E_1$ of the host (in light green) and virulence of its parasites (red line). Overall there are two energy types, $E = E_1$ and $E = E_2$, with two levels, $E^{lower}$ and $E^{upper}$, through which energy regulates the virulence of parasites that are sensitive to them. For simplicity, only one energy $E$ is shown ($E^{lower}$ dotted line in dark green, $E^{upper}$ dashed line in dark green). Whenever $E$ reaches its lower level virulence becomes maximal, and whenever $E$ attains its upper level virulence will be minimal. (A) Virulence is modelled as a step-function. (B) Virulence is modelled as a smooth-step function.

A

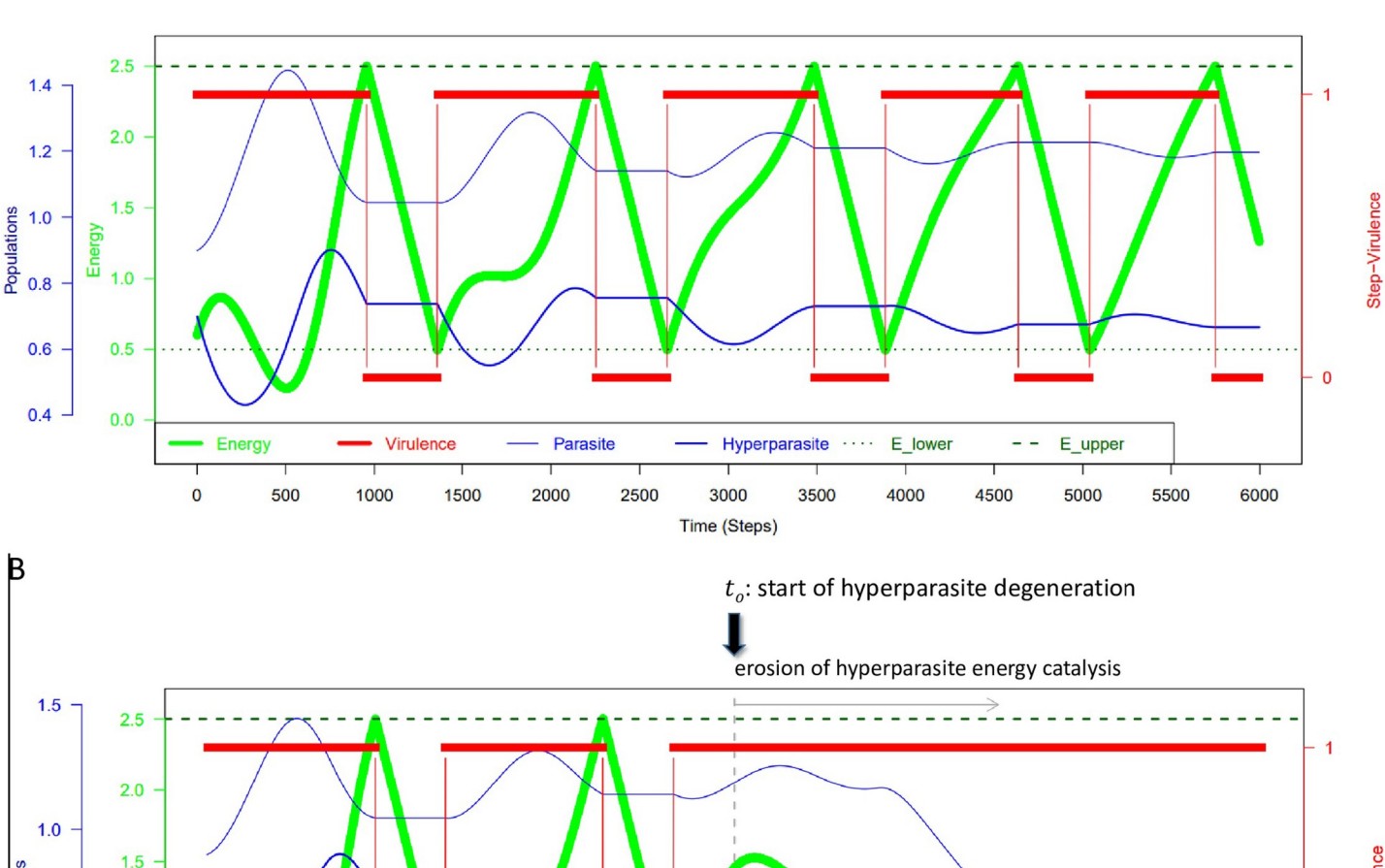

B

**Fig 2. One energy, one host-nested parasite unit homeostasis model.** (A) The model is extended by adding a hyperparasite $Y$ (phage, dark blue) to the parasite $X$ (bacteria, light blue) in order to form the host-nested parasite unit, limited to one single energy ($E = E_1$, in light green) that has two levels, $E^{lower}$ and $E^{upper}$ ($E^{lower}$ dotted line in dark green, $E^{upper}$ dashed line in dark green). (B) Hyperparasite degeneration is modeled starting at $t_0 \rightarrow t_1$ to illustrate the ensuing energy decline caused by the absence of newly generated catalytic energy.

$\beta > 0$ is the rate of energy consumption of parasite $X$ when virulent,

$\gamma > 0$ is the rate of catalytic energy production of hyperparasite $Y$ when virulent.

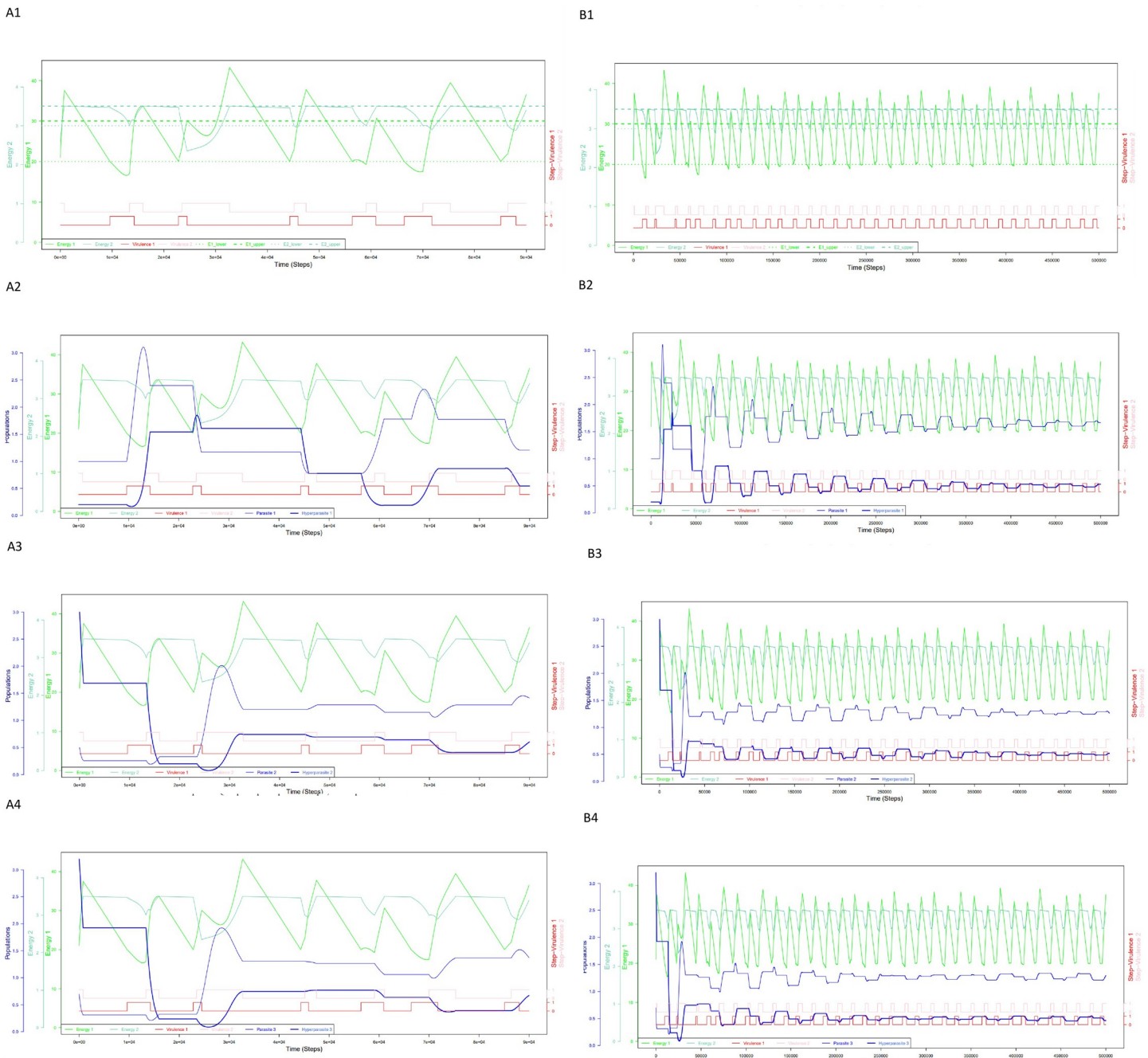

**Fig 3. Two energies, three host-nested parasite units homeostasis model.** (A1-A4) Represent the shorter (step 1, time 9e+04), and (B1-B4) the longer time scale (step 1, time 10e+05). The model is progressively extended by adding one, two and three host-parasite units ($X_{1-3}$, $Y_{1-3}$), and two energy levels ($E = E_1$, light green and $E = E_2$, dark green). The viru-lence of ($X_1$,$Y_1$) is regulated by the operating energy $E_1$ ($X_2$,$Y_2$) and ($X_3$,$Y_3$) are $E_2$– sensitive. (A1, B1) Show only two energies ($E_1$ and $E_2$) and two virulences ($\nu_{E1}$ in red, $\nu_{E2}$ in pink.(A2-A4, B2-B4) Show all three parameters simultaneously (population size in blue, parasites-hyper-parasites in light and dark blue, two energies and two virulences). (A2, B2) Display the parasite-hyperparasite pair ($X_1$, $Y_1$) responsible for $E_2 \rightarrow E_1$ transformation under virulence $\nu_{E1}$, (A3, B3) the parasite-hyperparasite pair ($X_2$, $Y_2$) responsible for $E_1$ catalysis under virulence $\nu_{E2}$, and (A4, B4) the parasite-hyperparasite pair ($X_3$, $Y_3$) responsible for $E_1 \rightarrow E_2$ transformation under virulence $\nu_{E2}$.

Other realizations would be that the parasite $X$ alone catalyzes the energy, or $X$ and $Y$ both catalyze the energy. It is assumed that after some time the hyperparasite $Y$ starts to degenerate

and loses its ability to catalyze new energy. This can be modelled as follows (Fig 2B), at some time $t_o$ the rate of catalytic energy production $\gamma$ starts to decline, $\dot{\gamma} = -d\gamma,\ t > t_0$, where $d$ is the decay rate of $\gamma$ or as we call it here, the degeneration rate of the phage $Y$. After $t_{halftime}$ = In (2)/$d$ the hyperparasite loses one half of its catalytic energy production power. Now, at some point in time, $t_1$, the host-nested parasite unit will consume more energy than the hyper-parasite $Y$ is able to catalyze. From this time on the host-nested parasite unit disintegrates. Since no energy-backup system is at hand in this model, the system rapidly breaks down with no time left to recruit a new, non-degenerate hyperparasite able to newly catalyze the necessary energy required for reestablishing homeostasis in the host-nested parasite unit (Fig 2B). Therefore, the "one energy, one host-nested parasite unit homomeostasis" exhibits a serious evolutionary in-stability.

As just pointed out, the elementary homeostasis model–based on one single energy $E = E_1$ regulating virulence and one host-nested parasite unit–exhibits considerably instability. In case the symbiotic parasite-hyperparasite pair lacks the ability to catalyze new energy $E$, for instance because of fitness erosion or an environmental change (see degenerative process just described above and Fig 2B), the system loses its problem-fixing ability and breaks down. In its problematic zone ($E < E^{lower}$), the system is in search mode, without obvious problem-fixing capacity. A new symbiotic prokaryote-virus pair capable of catalyzing new operating power $E$ has to be selected in the parasite pool from the habitat's available components. In this problematic zone, the system already has a low level of operating power $E$ that is further de-creasing, as long as no new solution is found. Accordingly, the system literally runs out of fuel until a new "fuel generator" can be identified. The inherently slow PRP needs to buy time in order to find a new, rapid solution. In conclusion, the system requires a second fallback energy system that is not available in the one energy model.

For this reason, we extend our one energy homeostasis model to an evolutionarily more stable two energies homeostasis model. In order to attain stable homeostasis using two energy levels ($E_1$, $E_2$), a quasispecies requires codifying the transformation of these energy forms into one another. This is accomplished by combining three individual host-nested parasite units ("two energies, three host-nested parasite units homeostasis model" in Fig 3). The differential equations for $E_1$ and $E_2$ are streamlined in the sense that parts of the energy $E_1$ consumption of parasites are now strictly separated from the energy transformation activities of the hyperparasites. In addition, the rate of energy consumption of the host now explicitly depends on the number of hosts $H$.

Fig 3 displays values of three different parameters in one picture, namely the population sizes of the parasites-hyperparasites, the energy levels, and virulence. In order to facilitate com-prehension, we deconvoluted these three groups of parameters in four distinct panels and two time scales, 9e04 for Fig 3A1–3A4 and 10e05 for Fig 3B1–3B4. Initially, in Fig 3A1 and 3B1, only two energies and two virulences are shown for simplicity. Subsequently, one, two and three parasite-hyperparasite pairs are introduced; Fig 3A2 and 3B2 focus on the ($X_1$, $Y_1$) parasite-hyperparasite pair responsible for the $E_2 \rightarrow E_1$ transformation under virulence $v_{E_1}$; Fig 3A3 and 3B3 focus on the ($X_2$, $Y_2$) parasite-hyperparasite pair responsible for the $E_1$ catalysis under virulence $v_{E_2}$, and Fig 3A4 and 3B4 focus on the ($X_3$, $Y_3$) parasite-hyperparasite pair responsible for the $E_1 \rightarrow E_2$ transformation under virulence $v_{E_2}$. Indeed, particularly Fig 3B2–3B4 show that the fluctuating population sizes of parasites and hyperparasites stabilize in time, slowly converging to an equilibrium. As can be seen, the $v_{E_2}$ virulence phases are always very short subperiods (Fig 3B2–3B4 and shown in pink), making the $E_1$ catalysis of ($X_2$, $Y_2$) and also the $E_1 \rightarrow E_2$ transformation of ($X_3$, $Y_3$) very rapid processes. This apparent accelera-tion may be inherent to the conditions modelled, or alternatively, truly reflect a biologically

relevant phenomenon. Higher energy conversion efficiency, this is what the observation suggests, indeed is associated with increasing cell size in prokaryotes and eukaryotes [35], and a primordial proto-cell having integrated three host-nested parasite units would formally meet this criterion.

Energy transformation by host-nested parasite units is supported by a vast body of literature, to name a few examples, i) the origin of the CRISPR-cas adaptive immune system in bacteria and archaea as a result of encounters with foreign, parasitic nucleic acids [36]; ii) the acquisition of adaptive immune V(D)J recombination in jawed vertebrates via parasitic *Proto-RAG* transposable elements [37]; iii) the emergence of 5-methylcytosine (5mC) in many species driven by the requirement to silence parasitic elements [38]; iv) programmed cell death (PCD) likely originating from antiparasitic defense mechanisms activated when immunity fails [39]; v) DNA parasites that led to the rapid, extensive intron gain during eukaryotic evolution [40], and finally, vi), the emergence of synaptic memory via Arc, a master regulator of synaptic plasticity that is related to a retrotransposon gene [41].

In conclusion, the main advantage of the "two energies homeostasis" over the "one energy homeostasis" lies in its robustness against environmental fluctuations and against functional defects of the symbiotic host-nested parasite units. If the system enters into a problematic zone, the slow process now has more time to find a new PRP, since this process relies on a second fallback energy.

A more in-depth consideration of the individual nature of the energy transformation steps is warranted, since energy transformation is a salient feature of the model. Notably, i) Process 2 associated with transforming $E_1 \rightarrow E_1$ is exemplified by the innate immune system [42]; ii) 5mC-methylation is a paradigm for Process 3 and $E_1 \rightarrow E_2$ transformation [24]; finally, iii) Process 1, implying $E_2 \rightarrow E_1$, is substantiated by acquired immunity and PCD [36, 37, 39], and by parasite-mediated cis-regulatory transcriptional regulation [43]. In support of an energy transformation paradigm, evidence for forth and back shuttling of parasite components between parasites and the host defense systems has recently been provided [44]. Moreover, the fact that fresh parasites are repeatedly recruited once ancient symbionts degenerated [12], a fate that appears to be the general rule for obligate endosymbionts [15], lends substantial credit to the idea of a host-nested parasite life cycle. We furthermore show that this host-parasite life cycle efficiently recycles energy from the fundamental process of replication-inherent parasites, whereby it captures, channels and transforms energy that can be used for further evolutionary purposes.

How can this life cycle of three host-nested parasite units then be integrated into a broader evolutionary genetic context? The core metrics of evolutionary genetics is fitness [45, 46]. We propose a model in ten steps centered on fitness management of a quasispecies. Step 1: a quasispecies faces an evolutionary problem (i.e. mutation, genetic drift, environmental change) when the range of Malthusian fitness values for all its members is negative (see Evolutionary problem solving; Methods). The problem resolution unfailingly implies an increase in fitness variance ($var(m)$). Step 2: a reduced fitness potentiates the environmental exposure, further decreases the mean fitness but increases the fitness variance. The increase in fitness variance in turn kick-starts the PRP, providing a potential solution. Step 3 specifies the counter forces mobilized by an eroded fitness, namely the two energy forms $E_1$ and $E_2$. Step 4 describes the fact that, in order to increase fitness, a quasispecies can only codify and enhance physical effects which are already in place (catalytic principle). Step 5: based on step 4 the quasispecies has merely problem solving solutions at hand that are inherent to its own replication system, namely parasites (catalytic principle). Those parasites subsequently codify and amplify the PRP in step 5. Parasites by their very nature gain in virulence whenever the host fitness is eroded, and parasite amplification enhances the process of fitness disintegration and lastly

incrementally boosts fitness variance. In step 6, this PRP amplification leads to recruitment of those elements among the parasitic waves that exhibit symbiotic potential; in other words, those symbionts codify and amplify the PRP. For simplicity, our quantitative simulations using LV-equations (LVhpr model) do not explicitly model the recruitment of new symbionts. But the simulations are compatible with the life cycle of symbionts. And the equations are such that they can be easily extended to modelling also the selection of new symbionts. However, the parasite population is kept stable–around an equilibrium–by the LV prey-predator-relationship bet-ween prokaryotes and their viruses. Step 7: therefore, parasites (or more precisely, prokaryote-phage pairs) qualify as catalysts, since i) the specific process is taking place in their absence, ii) if they are added the process is amplified, and iii) their population size is kept stable through-out and after the process. Step 7 implies that parasites are ubiquitous, and therefore, a host is a host-nested parasite unit. Prokaryote-phage pairs are mutually stabilized within the host-nested parasite unit, as shown by the LV-equations of the LVhpr model. Steps 8–10: the two afore-mentioned energies act as counterforce to parasitic virulence (step 8); for appropriate process stabilization the second, stored energy is required (step 9); finally and importantly (step 10), parasites do require to codify all relevant energy transformations, i.e. $E_2 \rightarrow E_1$, $E_1 \rightarrow E_1$ and $E_1 \rightarrow E_2$. Collectively, we established a ten-step model comprising a life cycle of three host-parasite units that build a catalytic metabolism. This metabolism captures, channels and trans-forms energy used by the host to resist the fitness erosion caused by mutation and genetic drift and to adapt to environmental stochasticity [30, 45, 47].

## Discussion

Our model capitalizes on the growing body of knowledge demonstrating the ubiquitous nature of genetic parasites and their inextricable ties to genome replication, and therefore to all life forms, including LUCA and pre-LUCA organisms. Furthermore, host-parasite coevolution is considered a major driving force of biological innovation and diversity.

This work first, confirms and extends previous findings by showing that parasites stabilize their host, modelling the host implicitly. Using LVhpr-extensions of the basic equations that integrate intra- and interspecies competition, we go on demonstrating that competition adds an addition layer of stability, i.e. smoothening the fluctuations (Fig 3B2–3B4). Second, it directly links–to the best of our knowledge for the first time–the central metabolic parameter, energy, with parasite abundance and host fitness. Third and most importantly, we provide a problem solving paradigm how adaptive processes involving host-nested parasite units linked to their energy resources and fitness potentially materialize and evolve, a core innovation. The coupling of the two processes involving problem solving and energy transformation is key. The initial host-parasite interactions in the context of the "one energy and one host-nested parasite unit" generally imply a very high exposure to symbiotic degeneration of the nested parasitism which can potentially lead to fast extinction. This first step allows a living system to continue evolving, however, still with life-threating degeneration processes. It then paves the way for more elaborate next moves, i.e. "two energies and three host-nested parasite units", which recruit synergies between hosts and parasites and allow the systems to evolve towards a more robust stability with less fluctuations.

One capital difference between our model and established hyperparasitism experiments [16] is the fact that we specifically select for conditions stabilizing the system via energy sources provided by the habitat. We therefore force mutualistic scenarios and do not explore host resistance factors that expose antagonistic forces and hyperparasites conferring hypervirulence [16]. Future work will therefore expand this model to address additional complexities,

including the effects exerted by direct and indirect anti-viral mechanisms such as virophages and antiviral responses, and their relevance for virulence and viral reservoirs.

An additional—admittedly relevant—limitation of our model is the fact that certain of its components, particularly the *catalytic principle*, invoke the second law of thermodynamics, the unifying cornerstone of physics [48], yet our model is not formulated throughout using the thermodynamic formalism, but centered on fitness. Indeed, numerous attempts have been made to apply thermodynamics to the origin of life and evolution [49, 50], and corresponding quantities in thermodynamics, machine learning and evolutionary biology were derived [49]. Moreover, statistical physics of open systems were formulated without assuming LUCA [51]. Applying this to the *catalytic principle*, in accordance with the maximum entropy production principle, the transformation rate along the path of least action becomes maximal, reframing the conventional view of enzymes acting across high-energy barriers [51]. No matter how relevant, a model purely based on thermodynamics is beyond the scope of the current manuscript and can be addressed in future studies.

## Supporting information

**S1 Fig. Host-nested parasite unit.** Parasite growth is unlimited (bacteria, phage), as long as the habitat nourishes it. Once the host loses the ability to nourish the parasites, they are condemned to perish. The association of a dedicated hyperparasite (phage) with a given parasite (bacteria) forming nested parasite pairs is a very effective and robust taming strategy for a parasitized host (both populations fluctuate around their equilibria).
(DOCX)

**S2 Fig. Catalytic host-nested parasite life cycle.** An initially stabilized host-nested parasite unit recruits symbionts (red) that catalyze energy transformation. Degenerate endosymbionts are replaced by fresh symbionts (green) that catalyze the next transformative round. A trio of such host-nested parasite units and two energies are required for adequate system stabilization and to build a catalytic life cycle.
(DOCX)

## Acknowledgments

We warmly thank Prof. Joseph Curran, University of Geneva, for critically reading the manuscript, and we are indebted to P. Norbert Widmer OSB† for his inspirational and fundamental insights.

## Author Contributions

**Conceptualization:** Bernard Conrad, Magnus Pirovino.

**Investigation:** Magnus Pirovino.

**Methodology:** Christian Iseli.

**Software:** Christian Iseli.

**Writing – original draft:** Bernard Conrad.

**Writing – review & editing:** Bernard Conrad.

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
