## [Decision Letter · Decision Letter 0]

8 Jan 2023

PONE-D-22-29899Energy-harnessing problem solving of primordial life: modeling the emergence of catalytic host-nested parasite life cycles

PLOS ONE

Dear Dr. Conrad,

Thank you for submitting your manuscript to PLOS ONE. After careful consideration, we feel that it has merit but does not fully meet PLOS ONE’s publication criteria as it currently stands. Therefore, we invite you to submit a revised version of the manuscript that addresses the points raised during the review process.

We look forward to receiving your revised manuscript.

Kind regards,

Jordi Villà-Freixa

Academic Editor

PLOS ONE

Journal Requirements:

Additional Editor Comments:

After careful selection of reviewers, only one that is suitable to provide useful criticism to your article has been submitted and considered by this editor. Agreeing with the comments of the referee, the article deserves publications in PLOS ONE because of its correct formulation although relatively obscure approximation to the issue that need to be addressed by the authors, following referee's comments.

Reviewers' comments:

Reviewer's Responses to Questions

**Comments to the Author**

1. Is the manuscript technically sound, and do the data support the conclusions?

Reviewer #1: Yes

2. Has the statistical analysis been performed appropriately and rigorously? 

Reviewer #1: N/A

3. Have the authors made all data underlying the findings in their manuscript fully available?

Reviewer #1: Yes

4. Is the manuscript presented in an intelligible fashion and written in standard English?

Reviewer #1: Yes

5. Review Comments to the Author

Reviewer #1: Review report of

PONE-D-22-29899

Energy-harnessing problem solving of primordial life: modeling the emergence of catalytic host-nested parasite life cycles

by Bernard Conrad, Christian Iseli, and Magnus Pirovino.

The paper presents a model of the energetic, replicative conditions of LUCA-like organisms and their parasites using Lotka-Volterra equations adapted to a nested parasite pair.

In my view, the paper could be published. Based on the stated assumptions, the model displays, by and large, the [damping oscillatory] behavior I expect.

So I am not questioning the work as such, only its relevance. I don’t think the LV model correctly presents energetics, i.e., thermodynamics, that drives evolution. We can do better. My point is substantiated by Mäkelä T, Annila A. Natural patterns of energy dispersal. Phys. Life Rev. 2010 7, 477-498. doi:10.1016/j.plrev.2010.10.001.

Sure, the Authors refer to the assumptions of the LV model, but still, in my view, the Readers would benefit from knowing that the thermodynamics of open systems describes abiogenesis without assuming LUCA, the hereditary material, etc. I am not insisting on this matter, only pointing out that mathematical modeling is not advancing our understanding of evolution, perhaps only obscuring it with model concepts and parameters that cannot be unambiguously related to the substance in evolution. For example, homeostasis is fine by corresponding to thermodynamic balance, while correspondence of the parasite capacity of the hosts in a given habitat to thermodynamics is vague.

Of course, one could argue that modeling is necessary to reduce the complexity to something practical, but in my view, not at the cost of comprehension. In thermodynamic terms, the oscillatory behavior results from free energy consumption that is large compared to the energy bound in populations. No question, the Authors’ model displays some conceivable scenarios, but it does not explain why things happen, causes and consequences, i.e., forces and changes in motion.

Yours sincerely,

Arto Annila

6. PLOS authors have the option to publish the peer review history of their article (what does this mean?). If published, this will include your full peer review and any attached files.

Reviewer #1: **Yes: **Arto Annila

---

## [Author Response · Author response to Decision Letter 0]

15 Jan 2023

Rebuttal letter for PONE-D-22-29899

Energy-harnessing problem solving of primordial life: modelling the emergence of catalytic host-nested parasite life cycles

Reviewer #1: Review report of PONE-D-22-29899

Energy-harnessing problem solving of primordial life: modelling the emergence of catalytic host-nested parasite life cycles by Bernard Conrad, Christian Iseli, and Magnus Pirovino.

The paper presents a model of the energetic, replicative conditions of LUCA-like organisms and their parasites using Lotka-Volterra equations adapted to a nested parasite pair.

In my view, the paper could be published. Based on the stated assumptions, the model displays, by and large, the [damping oscillatory] behavior I expect.

So I am not questioning the work as such, only its relevance. I don’t think the LV model correctly presents energetics, i.e., thermodynamics, that drives evolution. We can do better. My point is substantiated by Mäkelä T, Annila A. Natural patterns of energy dispersal. Phys. Life Rev. 2010 7, 477-498. doi:10.1016/j.plrev.2010.10.001.

Sure, the Authors refer to the assumptions of the LV model, but still, in my view, the Readers would benefit from knowing that the thermodynamics of open systems describes abiogenesis without assuming LUCA, the hereditary material, etc. I am not insisting on this matter, only pointing out that mathematical modeling is not advancing our understanding of evolution, perhaps only obscuring it with model concepts and parameters that cannot be unambiguously related to the substance in evolution. For example, homeostasis is fine by corresponding to thermodynamic balance, while correspondence of the parasite capacity of the hosts in a given habitat to thermodynamics is vague.

Of course, one could argue that modeling is necessary to reduce the complexity to something practical, but in my view, not at the cost of comprehension. In thermodynamic terms, the oscillatory behavior results from free energy consumption that is large compared to the energy bound in populations. No question, the Authors’ model displays some conceivable scenarios, but it does not explain why things happen, causes and consequences, i.e., forces and changes in motion.

As suggested by the Reviewer #1, we now fully acknowledge that a growing number of models were formulated in recent years aiming to apply thermodynamics to the origin of life and evolution, and dis-cuss the respective ramifications and implications (new refs. 49-51; swap of refs. 31 and 32).

Specifically, we added a new Materials and Methods section describing the catalytic principle and high-lighted its role in our model and relationship with the cornerstone second law of thermodynamics.

In addition, at the end of the discussion session we enumerate a number of recent models applying thermodynamics to this field. Specifically, as suggested by Reviewer# 1, we mention that statistical physics of open systems were developed without assuming LUCA (ref. 51). We furthermore also high-light how such open systems reframe the traditional views of enzymes (ref. 51), a concept that is cen-tral to the catalytic principle. 

In sum, we are convinced to have fully addressed the issues raised by Reviewer# 1.

---

## [Editor Report · Decision Letter 1]

30 Jan 2023

Energy-harnessing problem solving of primordial life: modeling the emergence of catalytic host-nested parasite life cycles

PONE-D-22-29899R1

Dear Dr. Conrad,

We’re pleased to inform you that your manuscript has been judged scientifically suitable for publication and will be formally accepted for publication once it meets all outstanding technical requirements.

Kind regards,

Jordi Villà-Freixa

Academic Editor

PLOS ONE
---

## [Editor Report · Acceptance letter]

6 Feb 2023

PONE-D-22-29899R1 

Energy-harnessing problem solving of primordial life: modeling the emergence of cata-lytic host-nested parasite life cycles 

Dear Dr. Conrad:

I'm pleased to inform you that your manuscript has been deemed suitable for publication in PLOS ONE. Congratulations! Your manuscript is now with our production department. 

Kind regards, 

on behalf of

Dr. Jordi Villà-Freixa 

Academic Editor

PLOS ONE